# Beeswax Alcohol and Fermented Black Rice Bran Synergistically Ameliorated Hepatic Injury and Dyslipidemia to Exert Antioxidant and Anti-Inflammatory Activity in Ethanol-Supplemented Zebrafish

**DOI:** 10.3390/biom13010136

**Published:** 2023-01-09

**Authors:** Youngji Han, Seonggeun Zee, Kyung-Hyun Cho

**Affiliations:** 1Raydel Research Institute, Medical Innovation Complex, Daegu 41061, Republic of Korea; 2LipoLab, Yeungnam University, Gyeongsan 38541, Republic of Korea

**Keywords:** beeswax alcohol (BWA), fermented-black rice bran (BRB-F), alcohol-induced liver injury, dyslipidemia, high-density lipoproteins, interleukin-6

## Abstract

Alcohol abuse, a global health problem, is closely associated with many pathological processes, such as dyslipidemia and cardiovascular disease. In particular, excessive alcohol consumption promotes dyslipidemia and liver damage, such as hepatic steatosis, fibrosis, and cirrhosis. Beeswax alcohol (BWA) is a natural product used for its antioxidant properties that has not been evaluated for its efficacy in alcohol-induced liver injury. In the present study, zebrafish were exposed to 1% ethanol with supplementation of 10% fermented black rice bran (BRB-F), 10% BWA, or 10% mixtures of BWA+BRB-F (MIX). The BRB-F, BWA, and MIX supplementation increased the survival rate dramatically without affecting the body weight changes. In histology of hepatic tissue, alcoholic foamy degeneration was ameliorated by the BWA or MIX supplements. Moreover, dihydroethidium (DHE) and immunohistochemistry staining suggested that the MIX supplement decreased the hepatic ROS production and interleukin-6 expression significantly owing to the enhanced antioxidant properties, such as paraoxonase. Furthermore, the MIX supplement improved alcohol-induced dyslipidemia and oxidative stress. The BWA and MIX groups showed lower blood total cholesterol (TC) and triglyceride (TG) levels with higher high-density lipoprotein-cholesterol (HDL-C) than the alcohol-alone group. The MIX group showed the highest HDL-C/TC ratio and HDL-C/TG ratio with the lowest low-density lipoprotein (LDL)-C/HDL-C ratio. In conclusion, BWA and BRB-F showed efficacy to treat alcohol-related metabolic disorders, but the MIX supplement was more effective in ameliorating the liver damage and dyslipidemia, which agrees with an enhanced antioxidant and anti-inflammatory activity exhibited by BWA/BRB-F in a synergistic manner.

## 1. Introduction

Chronic heavy alcohol consumption is intimately associated with alcohol-related liver disease (ALD), which contributes to liver injury, ranging from hepatic steatosis to hepatitis and hepatic cirrhosis [1]. Excessive accumulation of the metabolic end products from alcohol metabolism can cause oxidative stress, lipid peroxidation, and inflammation and promote fat accumulation [2]. Oxidative stress and inflammation are major secondary factors that promote liver injury [2]. According to the World Health Organization (WHO), in 2016, there were more than 3 million deaths (5.3% of all deaths) annually from alcohol use globally [3]. Moreover, alcohol is responsible for 14.5% of all disability-adjusted life years [4]. In Korea, 4809 Koreans died from alcohol-induced causes, as reported by Statistics Korea in 2017 [5]. Therefore, industrial and personal interest in health functional foods to improve alcohol-induced metabolic diseases is increasing.

Beeswax alcohol (BWA), a substance purified from beeswax, contains a mixture of six primary aliphatic alcohols (C24, C26, C28, C30, C32, and C34), and has been reported to have antioxidant, anti-platelet, cholesterol-lowering, and gastroprotective effects [6,7]. In particular, BWA supplementation promoted the secretion and quality of gastric mucus in ethanol-ulcer rat models and attenuated inflammation in antigen-induced arthritis [8,9]. With these properties, BWA has been registered as a functional food ingredient for joint and gastrointestinal health in Korean health functional foods by the KFDA.

Fermented black rice bran (BRB-F) has antioxidative, anti-inflammatory, anti-allergic, and anticarcinogenic effects with high contents of phenolics, flavonoids, and anthocyanin [10,11,12]. BRB-F with glutathione-enriched yeast extract improved alcohol-induced hangovers [13]. BRB-F contains polysaccharides, which have been reported to have various physiological activities, protection against necrosis of the liver, inhibition of *Salmonella typhimurium* infection, and inhibition of endotoxemia by activating the Th1 immune response in vivo [14]

A few natural products have shown adequate protection against alcohol-induced injuries and ameliorate hangover symptoms in animals and humans [15]. On the other hand, there are limited natural products registered as health functional foods for alcohol-induced injuries and hangovers in the Republic of Korea. Although it has been reported that BWA ameliorated indomethacin-induced gastric ulcers [8], there has been no report about the efficacy of BWA on ethanol-related liver damage.

To investigate the physiologic effect of a mixture of BRB-F and BWA on vertebrate animals, we employed a zebrafish (*Danio rerio*) model, in which dyslipidemia and liver damage were induced by ethanol exposure as suggested previously [16,17,18,19]. By exposing adult zebrafish to ethanol (final 0.5–2% in system water) for several days, liver damage was induced, including dyslipidemia [16], elevation of serum triglyceride (TG), alanine aminotransferase (ALT) [17], oxidative stress [18], and inflammation [19].

However, on the other hand, no studies have been reported to evaluate the improvement of alcoholic disease by supplementation of BRB-F, BWA, and co-supplementation of BRB-F and BWA. Therefore, in the current study, BWA, BRB-F, and a mixture of BWA and BRB-F (MIX) were tested to ameliorate alcohol-induced liver injury in zebrafish. The present study evaluated the efficacy of 10% BWA, 10% BRB-F, and 10% MIX supplement on alcohol-induced liver injury in zebrafish exposed to 1% ethanol in system water.

## 2. Materials and Methods

### 2.1. Materials

The BWA was obtained from Rainbow and Nature Pty, Ltd. (Thornleigh, NSW, Australia). The material contained six high-molecular-weight alcohols purified from beeswax (from *Apis mellifera*, L.) with the following composition: tetracosanol (6–15%), hexacosanol (7–20%), octacosanol (12–20%), triacontanol (25–35%), dotriacontanol (18–25%), and tetratriacontanol (≤7.5%) (purity ≥ 85%) [20].

The BRB-F was obtained from STR Biotech (Chuncheon, Gangwon-do, Republic of Korea), and the material is manufactured as described previously [13,14] with slight modification. Briefly, BRB-F was made from a liquid medium of black rice bran. The marker ingredient of BRB-F is γ-oryzanol, and it is contained at a concentration of about 100 mg/kg.

The BRB, BWA, and MIX powder, each 100 mg of powder, was dissolved in 10 mL of ethanol, with continuous agitation for 12 h at room temperature. Then, the extract was centrifuged (3000× *g*) to pellet down any debris. The supernatant was diluted with ethanol to obtain 1, 2, 5, 10 mg/mL and tested for the in vitro assay.

### 2.2. DPPH-Radical-Scavenging Assay and Ferric-Ion-Reducing Ability

A solution of diphenyl-1-picrylhydrazyl (DPPH) free radicals was prepared by dissolving 2.4 mg of DPPH in 100 mL methanol using the standard method [21]. The DPPH solution (0.95 mL) in ethanol was mixed with the BRB-F, BWA, and MIX as a source of antioxidants. The mixture was observed continuously at 517 nm for 60 min at 25 °C using a UV-2600i spectrophotometer (Shimadzu, Kyoto, Japan) with LabSolutions software UV-Vis 1.11 (Shimadzu, Kyoto, Japan). The percentage of inhibition against the DPPH radical was calculated with the following equation to report the free-radical-scavenging activity (FRSA).
FRSA (% inhibition) = % inhibition = ((Ac − As)/As)) × 100
where Ac is the absorbance of the control and As is the absorbance of the tested sample or working standards.

The ferric-ion-reducing ability (FRA) was determined using the method reported by Benzie and Strain [22]. Briefly, the FRA reagents were freshly prepared by mixing 0.3 M sodium acetate buffer (pH 3.6), 2.5 mL of 10 mM 2,4,6-tripyridyl-*S*-triazine, and 2.5 mL of 20 mM FeCl_3_·6H_2_O. The antioxidant capacities of BRB-F, BWA, and MIX were estimated by measuring the increase in absorbance induced by the ferrous ions generated. Freshly prepared FRA reagent (300 µL) was mixed with extracts of BRB-F, BWA, and MIX as an antioxidant source. The FRA was then determined by measuring the absorbance at a wavelength of 593 nm using a UV-2600i spectrometer (Shimadzu, Kyoto, Japan) with LabSolutions software UV-Vis 1.11 (Shimadzu, Kyoto, Japan).

### 2.3. Zebrafish Maintenance

All experiments were performed using the zebrafish auto system (Genomic Design Bioengineering Company; GDBC, Daejeon, Republic of Korea). The water temperature was kept at 28 ± 1 °C. The photoperiod was 14 h of light and 10 h of dark. Zebrafish over 16 weeks old were divided randomly into five groups; each group had 60 zebrafish exposed to 1% ethanol and supplemented with one of the experimental diets in zebrafish diet (Tetrabit, Gmbh D49304, Melle, Germany) for 17 days. The groups were as follows: exposed to 1% PBS group (PBS, phosphate buffered saline, control, n = 60), exposed to 1% ethanol group (EtOH, fed normal diet, n = 60), BRB-F (exposed to 1% ethanol, fed a normal diet with 10% fermented black rice bran, wt/wt, n = 60), BWA (exposed to 1% ethanol, fed a normal diet with 10% beeswax alcohol, wt/wt, n = 60), mixture of BRB-F and BWA (MIX, exposed to 1% ethanol, fed a normal diet with 10% mixture of BRB-F and BWA (6:1), wt/wt, n = 60). Before feeding and ethanol exposure, all groups were acclimated to each diet for one week. Subsequently, the zebrafish were exposed to 1% ethanol except for the PBS control group and fed twice daily at 9 am and 6 pm with 10 mg of designated diet per zebrafish.

The system water containing 1% ethanol solution (*v*/*v*) in tank was newly replaced every 24 h, and dead fish were monitored and collected daily at 9 a.m. The survival rate was measured during the test duration of up to 17 days post-exposure.

After feeding, blood (2 µL) was drawn from the heart of adult zebrafish and combined with 4 µL of PBS-EDTA, then collected into EDTA-treated tubes (a final concentration of 1 mM) as per our previous report [23]. Plasma of each group was collected after centrifugation (5000× *g*) for 15 min.

### 2.4. Plasma Lipid Profile

The plasma total cholesterol (TC), high-density lipoprotein cholesterol (HDL-C), and triglyceride (TG) levels were determined using commercially available assay kits (Asan Pharmaceutical, Hwasung, Republic of Korea). In addition, aspartate transaminase (AST), alanine transaminase (ALT), and gamma-glutamyl transferase (GGT) were measured using a commercially available assay kit (Asan Pharmaceutical, Hwasung, Republic of Korea).

### 2.5. Ferric-Ion-Reducing Ability of Plasma

The ferric-ion-reducing ability of plasma (FRAP) was measured using the method reported [21]. It was determined by measuring the absorbance at 593 nm using a microplate reader (BioRad iMark MicroPlate Reader; Bio-Rad, Hercules, CA, USA).

### 2.6. Paraoxonase Activity

The paraoxonase (PON) activity was measured using a slight modification of the method reported by Mackness et al. to measure the increase in the absorbance of *p*-nitrophenol [24]. It was determined by measuring the absorbance at a wavelength of 415 nm using a microplate reader (BioRad iMark™ MicroPlate Reader; Bio-Rad, Hercules, CA, USA).

### 2.7. Histological and Immunohistochemical Analysis

The hepatic tissue was removed from the zebrafish, fixed in a 10% formalin buffer solution, and frozen. Fixed liver tissue was processed routinely for paraffin embedding, and a 7 μm thick section was prepared and stained with Hematoxylin and Eosin (H&E) [25], as well as immunohistochemistry (IHC) [26], and observed under an optical microscope (Nikon, Tokyo, Japan), with a magnification of ×400. In the IHC staining for IL-6, the primary antibody was diluted according to the manufacturer’s instructions and incubated overnight at 4 °C (1:200; ab9324, Abcam, London, UK). The IHC reaction was visualized using EnVision + System-HRP polymer kit as a secondary antibody (1:1000, Code K4001, Dako, Denmark).

### 2.8. Imaging of Reactive Oxygen Species

Frozen hepatic tissues were stained with dihydroethidium (DHE, 37291, Sigma, St. Louis, MO, USA) to visualize the reactive oxygen species (ROS) levels, as described previously [27]. The image of ROS was obtained by fluorescence observation (Ex = 585 nm and Em = 615 nm) via fluorescence microscopy using a Motic PA53 FS6 (Motic, Hong Kong SAR, China).

### 2.9. Statistical Analysis

The data are expressed as the mean ± standard error of mean (SEM). All statistical analyses were performed using IBM SPSS Statistics 26 (SPSS, Inc., Chicago, IL, USA). One-way analysis of variance (ANOVA) and Tukey’s multiple comparisons tests were carried out to test for any significant differences between the means (*p* < 0.05). Correlation analysis between TC, TG, HDL-C, and IL-6 stained area was carried out with Spearman correlation analysis using the IBM SPSS. A *p*-value < 0.05 was considered significant.

## 3. Results

### 3.1. Mixture of BWA and BRB-F Showed the Strongest Antioxidant Activities in a Synergistic Manner

The DPPH-radical-scavenging assay and FRA revealed the potent antioxidant ability of BRB-F, BWA, and MIX (Figure 1). As shown in Figure 1A, the MIX treatment showed the highest scavenging activity of DPPH radical, up to a 58% reduction from the initial level at final 10 mg/mL, while treatment with BRB-F and BWA showed 28% and 35% reductions, respectively at final 10 mg/mL. The MIX treatment at 2 and 5 mg/mL (final) showed a similar extent of the radical-scavenging activity with vitamin C (Vit-C) treatment at 12 and 28 µM (final). Hence, the MIX had the strongest antioxidant ability with a synergistic manner.

In the FRA assay, as shown in Figure 1B, the BWA showed a lower ferric-ion-reducing ability than its DPPH-removal ability because BWA is insoluble in an aqueous buffer system, although BWA showed a 9% higher ability than the EtOH-alone control. However, the MIX showed the highest FRA, indicating that the solubility of BWA was enhanced in the mixture with BRB-F. The MIX showed the highest antioxidant activity in both the DPPH radical removal assay and the FRA assay in a synergistic manner.

### 3.2. Mixture of BWA and BRB-F Increases the Survival of Zebrafish under Supplementation with 1% EtOH

As shown in Table 1, the summarized experimental design and body-weight changes during the experiments were described.

The survival of the PBS control group did not change during the 17-day feeding period (Figure 2A). A trend towards remarkably lowered survival in zebrafish exposed to 1% ethanol was observed. Until day 10, the EtOH-alone group showed 80% survival. All the zebrafish in the EtOH group died by the 11th day of feeding. On the other hand, the BRB-F, BWA, and MIX supplement groups showed increased survival compared to the 1% EtOH-alone group. The MIX and BWA groups showed the highest survival, up to 97–98%, while the BRB-F group showed 91% survival.

Before 1% ethanol exposure, there was no significant difference in the body wet weights among the groups (Figure 2B,C). After 10 days of 1% ethanol exposure, the body wet weight of the EtOH group decreased by 10% compared to the PBS control group, although there was no significance. On the other hand, there was no significant difference in the body wet weights in the ND, BRB-F, BWA, and MIX groups. After 10 days of exposure to 1% ethanol, the wet body weights in the all groups were similar to the values at day 0 of this study (Table 1).

### 3.3. Mixture of BWA and BRB-F Improves the Alcohol-Induced Liver Injury in Zebrafish under Supplementation with 1% EtOH

After 10 days of 1% ethanol exposure, H&E staining of the hepatic tissue showed that the EtOH-alone group had a significantly larger stained area of the nucleus (up to 14%) than the PBS control group (Figure 3), indicating that ethanol exposure caused infiltration of neutrophils in hepatocytes. In addition, acute alcoholic foamy degeneration around the portal vein was detected in the EtOH-alone group (black arrow in Figure 3). The MIX group showed the lowest H&E stained area, 8% lower than the EtOH-alone group. Although there was no significant difference between the BRB-F and BWA groups, the MIX group showed a lowered H&E stained area and no sign of alcoholic foamy degeneration compared to the BRB-F and BWA groups. Hence, the BWA and BRB-F mixture could protect zebrafish from alcoholic liver damage synergistically.

### 3.4. Mixture of BWA and BRB-F Suppressed ROS Production with Anti-Inflammatory Properties in Zebrafish under Supplementation with 1% EtOH

DHE staining showed that the EtOH-alone group had the strongest red intensity, approximately two-fold higher than the PBS-alone group, indicating the highest ROS production (Figure 4). On the other hand, the BRB-F and BWA groups showed a trend toward a decreased DHE-stained area, up to 37% and 28%, respectively, lower than the ethanol-alone group. Moreover, the MIX group showed the lowest ROS production: 45% lower than the EtOH-alone group. IHC analysis with the IL-6 antibody revealed that hepatic IL-6 expression was stimulated by EtOH exposure (Figure 4). After 10 days of exposure to 1% ethanol, the EtOH group showed a significant increase in IL-6 expression (32%) than the PBS control group. In contrast, the BRB-F, BWA, and MIX supplements decreased those values in the EtOH group by up to 14%, 19%, and 23%, respectively. Hence, MIX supplementation could suppress alcohol-induced hepatic ROS production and acute inflammation.

A comparison of hepatic injury biomarkers, plasma AST, ALT, and GGT groups showed that (Figure 5) the EtOH-alone group showed the highest plasma AST, ALT, and GGT: 1.9-, 2.1-, and 1.6-fold higher than the PBS control group. The MIX group showed the lowest ALT and GGT levels, around a similar level to the PBS control group. Interestingly, the BRB-F, BWA, and MIX groups showed similar serum AST levels to the PBS control group. The MIX supplementation ameliorated all the hepatic injury biomarkers to a similar level to the PBS control group.

The plasma FRA in the EtOH group was significantly lower than the PBS control group (Figure 6A). The BRB-F, BWA, and MIX groups significantly increased the FRA: 5%, 25%, and 46%, respectively, compared to those values of the EtOH group. PON activity was decreased by EtOH exposure, 57% lower than that of the PBS control group, indicating that heavy ethanol consumption impaired the antioxidant activity of blood. However, the BRB-F, BWA, and MIX groups showed 1.5-fold, 1.6-fold, and 2.2-fold, respectively, higher PON activity than the ethanol control group.

### 3.5. Mixture of BWA and BRB-F Attenuates the Plasma Lipid Profiles in Zebrafish under Supplementation with 1% EtOH

During the 10 days post-exposure, the changes in lipid profile are shown in Figure 7. After 10 days of ethanol exposure, plasma HDL-C levels in the EtOH group were approximately 2.1-fold lower than the PBS control group. The BRB-F, BWA, and MIX group showed significantly higher plasma HDL-C levels compared to those of the EtOH group. The plasma HDL-C/TC ratio (%) of the EtOH-alone group (14.6%) was 56% lower than the PBS-alone group (32.9%), indicating a decrease in HDL-C by ethanol consumption. However, the BRB-F, BWA, and MIX groups showed 17.5%, 21.5%, and 22.6%, respectively, of HDL-C/TC, suggesting that the dyslipidemia could be improved by the extracts, especially by the MIX.

Moreover, the plasma TG levels were approximately increased two-fold by ethanol exposure, indicating that alcoholic liver damage was associated with hypertriglyceridemia. On the other hand, the BRB-F, BWA, and MIX groups showed 30%, 41%, and 47% lower plasma TG levels, respectively. The HDL-C/TG ratio was significantly lower in the EtOH group than in the PBS control group (~8% vs. 36.2%). The BRB-F, BWA, and MIX groups showed HDL-C/TG ratios of 13, 21, and 21%, respectively.

After 17 days of 1% ethanol exposure, all the zebrafish in the EtOH group died. The plasma TC and TG levels were significantly lower in the MIX group than in the BRB-F and BWA group (Table 2). Interestingly, MIX supplementation resulted in a two-fold decrease in the plasma TG levels than the BRB-F group. Furthermore, the MIX group showed the lowest plasma AST, ALT, and GGT levels, up to 37 units lower than the EtOH-alone group.

## 4. Discussion

The current study is the first test to evaluate the efficacy of BWA, BRB-F, and MIX on liver injury induced by heavy alcohol consumption and dyslipidemia. In the DPPH-radical-scavenging assay and FRA assay, BRB-F and BWA exhibited potent antioxidant capacity synergistically. MIX containing BRB-F and BWA at a 1:6 ratio (*w*/*w*) enhanced the antioxidant capacities and solubility. Therefore, it was possible to postulate that the MIX has a higher bioavailability against alcohol-induced dyslipidemia and hepatic injury than BRB-F and BWA alone.

Zebrafish have been used as chronic and acute alcohol-induced liver injury models [28,29]. Ethanol was added to the water (final 1%) and maintained at that level every 24 h to induce hepatic steatosis and fibrogenic responses in the zebrafish larvae [30]. A previous study reported that all zebrafish died after 10 days of 1% ethanol (*v*/*v*) exposure [16]. In the present study, in accordance with previous studies, adult zebrafish were maintained in a 1% ethanol solution to develop an alcohol-induced liver injury model. Consistent with the previous study [16], all zebrafish died after 10 days in the ethanol-alone group, but the BRB-F, BWA, and MIX groups showed much higher survival until sacrifice at 17 days post-exposure. Although BWA contains six primary aliphatic alcohols, C24–C34, which also can be oxidized by alcohol dehydrogenase (ADH), the oxidation of BWA may not increase the ethanol-induced toxicity due to having a much lower amount than EtOH. Indeed, the final concentration of 1% EtOH in the water tank was equal to 172 mM EtOH in the water tank for 24 h exposure, while 1 mg of BWA in the diet for 300 mg of zebrafish (Table 1) corresponded to around 4.8 mM in the zebrafish. Because the amount of EtOH exposure was 36-fold higher than the consumption of BWA, it is plausible to postulate that the oxidation of EtOH by ADH was more predominant than that of BWA, especially in competitive inhibition mode of ADH.

ALD, from mild disease to alcoholic hepatitis and cirrhosis, are some of the most common causes of morbidity and mortality globally [31]. Ethanol consumption produces excessive ROS, impairs antioxidant properties, elevates oxidative stress, and increases inflammatory cytokines in the liver, frequently associated with dyslipidemia [32]. In the persistent condition, alcoholic liver injury contributes to alcoholic hepatitis and cirrhosis [33]. In morphological analysis, alcoholic foamy degeneration usually occurs without hepatic steatosis and fibrosis despite chronic alcohol consumption [32]. Consistent with these results, alcoholic foamy degeneration in the EtOH group was observed in the hepatic H&E staining analysis. On the other hand, however, BRB-F, BWA, and MIX improved the alcoholic foamy degeneration (Figure 3) with less ROS and IL-6 production in the liver. The consumption of beeswax alcohol improved lipid peroxidation in middle-aged and old subjects [34]. BRB-F has been famous for its antioxidant activity with high anthocyanin and γ-oryzanol content [10,13,14]. In the same context, BRB-F, BWA, and MIX supplementation lowered ROS production in hepatic tissue (Figure 4) with increased FRA and PON activity in zebrafish plasma (Figure 6) Overall, the MIX ameliorated alcohol-induced hepatic inflammation with high antioxidant properties.

An elevation of the IL-6 level in blood and hepatic tissue is associated with the progression of alcoholic liver injury in animal and human models [35]. IL-6 is notorious for a proinflammatory cytokine that stimulates hsCRP production during acute inflammation and infection [36]. IL-6 was elevated during the acute inflammatory response, which promotes the transition to progressive hepatic damage and acute phase [37]. In the current study, elevated hepatic IL-6 expression caused by ethanol exposure was decreased significantly in the BWA and MIX groups (Figure 4). Furthermore, blood biomarkers of liver function, such as AST, ALT, and GGT, were decreased by supplementation of BRB-F, BWA, or MIX in a synergistic manner. These results suggested that the consumption of BRB-F, BWA, or MIX attenuated the alcohol-related hepatic damage along with reducing IL-6 expression and the index of oxidative damage. In addition, IL-6 secretion is promoted by chronic alcohol consumption, and is closely related to the affective symptoms in alcohol-dependent patients [38]. These findings suggested that the MIX might have a tremendous anti-hangover ability compared to BRB-F alone because it has a more significant effect against alcohol-induced liver injury synergistically.

Heavy alcohol consumption is frequently associated with lipid metabolism disorder [39,40]. The concentration of TG and the LDL-C/HDL-C ratio were increased significantly in middle-aged alcohol drinkers who consumed more than 30 g of alcohol per day [41]. Long-term alcohol consumption in middle-aged women, even in small amounts, caused a significant decrease in the serum HDL-C and apoA-I with atherogenic changes in the LDL and HDL, such as an increase in the TG and malondialdehyde (MDA) contents with a loss of PON activity [42]. Taken together, chronic alcohol consumption is closely linked with incidence of dyslipidemia and atherosclerosis.

The current study also showed that 1% ethanol exposure caused severe dyslipidemia, increased the plasma TG levels, and decreased the HDL-C levels and % HDL-C/TC. The BRB-F, BWA, or MIX supplementation increased the HDL-C levels and % HDL-C/TC, and decreased the plasma TG levels. In particular, the BWA group showed the highest HDL-C level (~47 mg/dL), while the BRB-F and MIX groups showed 35 and 33 mg/dL, respectively. Alcohol-induced dyslipidemia, especially lower HDL-C and higher TG, is linked to an elevation of inflammatory cytokines, such as IL-6. There are positive correlations between serum levels of TG and IL-6 with weight regain and fat mass expansion [43]. The decrease in blood TG is directly associated with an amelioration of inflammatory response in the acute phase because intravenous administration of IL-6 stimulated secretion of TG in blood with the highest level at 2 h post-injection in a dose-dependent manner [44]. As shown in Appendix A, a correlation analysis revealed that the plasma TG level was positively correlated (*r* = 0.039, *p* < 0.001) with the IL-6 stained area in hepatic tissue. Taken together, previous reports and the current results show a good agreement that the elevation of IL-6 is associated with an increase in blood TG.

On the other hand, low HDL-C is associated with high levels of IL-6 in the plasma (*r* = −0.23, *p* < 0.01) and TG level (*r* = −0.44, *p* < 0.01) in a cohort of older adults, 65–102 years old, from the InChianti study [45], indicating that HDL-C is inversely correlated with the inflammatory parameters. Furthermore, human plasma HDL and reconstituted HDL inhibited IL-6 production in human endothelial cells in a concentration-dependent manner by inhibiting p38 MAP kinase [46]. HDL downregulated the IL-6 mRNA levels, and subjects with low HDL-C showed significantly elevated plasma IL-6 levels [47]. IL-6 upregulated the synthesis of acute-phase proteins in hepatocytes, such as high-sensitive C-reactive protein (hsCRP) [48]. In the current study, the plasma HDL-C level was negatively correlated (*r* = −0.646, *p =* 0.043) with the IL-6 stained area in hepatic tissue (Appendix A). These results suggest that elevation of HDL-C is closely linked with suppression of inflammatory cytokines and signaling cascade.

Previous studies suggested that fermented BRB-F increased the antioxidant and antimicrobial activity regardless of the microbial species [49,50]. In addition, a combination of BRB-F and glutathione-enriched yeast effectively protects against alcohol-induced hangovers in mice [51]. On the other hand, BWA has gastroprotective and antioxidant properties by reducing lipid peroxidation effectively, which helps improve inflammation in osteoarthritis in rats and human subjects [52,53,54,55,56]. Based on these findings, a mixture of BWA and BRB-F has higher antioxidant properties than each alone. Moreover, co-administration between BWA and grape seed extract showed synergistic effects as antioxidants in human subjects [57].

The main component of BRB-F is γ-oryzanol [13,14,58] and the major ingredients of BWA are six long-chain aliphatic alcohols from C24 to C34 [20,52,53]. Because the γ-oryzanol and the six long-chain aliphatic alcohols are very hydrophobic, we can postulate a putative interaction of structural compliance between γ-oryzanol and long-chain aliphatic alcohols via hydrophobic interactions. Since γ-oryzanol and BWA are lipophilic antioxidants, their synergistic effects might originate from the formation of a critical micelle concentration in the non-polar phase, as suggested previously [59]. Furthermore, there have been reports that show a blood-lipid-lowering effect of both γ-oryzanol and BWA in humans [60,61]. The calculated IC_50_ from the regression analysis of the FRSA (% inhibition) [62] from Figure 1 were 19.5 mg/mL, 14.8 mg/mL, and 7.2 mg/mL for BRB-F, BWA, and MIX, respectively, indicating a putative synergistic effect between BWA and BRB-F. From the same FRSA analysis, the IC_50_ of Vitamin C was calculated as 23.5 mM, which is very similar with other reports of around 19.8–34.5 mM [63,64]

Taken together, it is possible to expect of synergistic effect of BWA and BRB-F to improve dyslipidemia via the combined antioxidant activities of both BWA and BRB-F in vitro and in vivo.

## 5. Conclusions

These findings proved the potent amelioration effects of BRB-F, BWA, and MIX on alcohol-induced hepatic damage and dyslipidemia in adult zebrafish via antioxidant and anti-inflammatory activity. The MIX group showed the strongest protective effects to suppress inflammation and ROS production in the liver and blood with enhanced PON activity in a synergistic manner, compared with the BRB-F group and BWA group. The MIX supplementation showed better results than BWA and BRB-F alone, and increased the survival of the zebrafish by suppressing hepatic inflammation and improving lipid profiles, raising the %HDL-C in TC, and lowering TG level. Future research will be necessary to investigate lower concentrations of BRB-F, BWA, and MIX in the same experiment to find an optimum dose of the extract to maximize the efficacy.

## Figures and Tables

**Figure 1 biomolecules-13-00136-f001:**
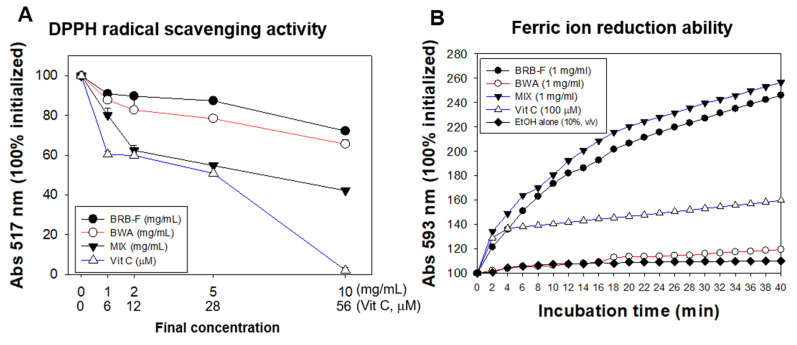
Antioxidant ability of BRB-F, BWA, and MIX in comparison with vitamin C (Vit-C). (**A**) DPPH-radical-scavenging activity. (**B**) Ferric-ion-reduction ability; BRB-F, fermented black rice bran; BWA, beeswax alcohol; MIX, 1 mg/mL mixture of BWA and BRB-F (1:6, *w*/*w*); DPPH, 2,2-diphenyl-1-picrylhydrazyl.

**Figure 2 biomolecules-13-00136-f002:**
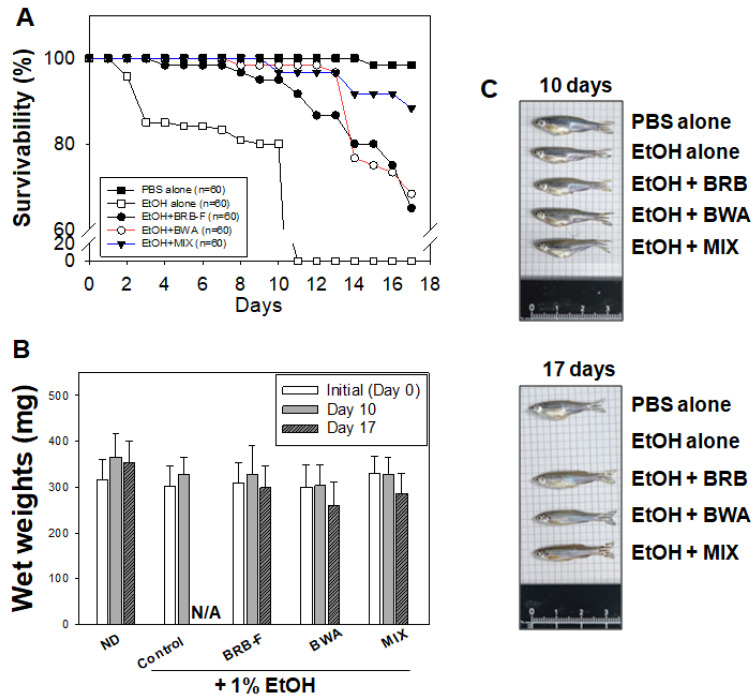
Effect of BRB-F, BWA, and MIX in alcohol liver injury zebrafish for 17 days on (**A**) survival and (**B**) body weights; PBS, normal diet, n = 20; EtOH, zebrafish exposed to 1% ethanol, n = 20; BRB-F, EtOH fed 10% black rice bran, n = 20; BWA, EtOH fed 10% beeswax alcohol, n = 20; MIX, EtOH fed a mixture of BWA and BRB-F (1:6, *w*/*w*), n = 20. Photos of zebrafish body at 10 days and 17 days are shown (**C**).

**Figure 3 biomolecules-13-00136-f003:**
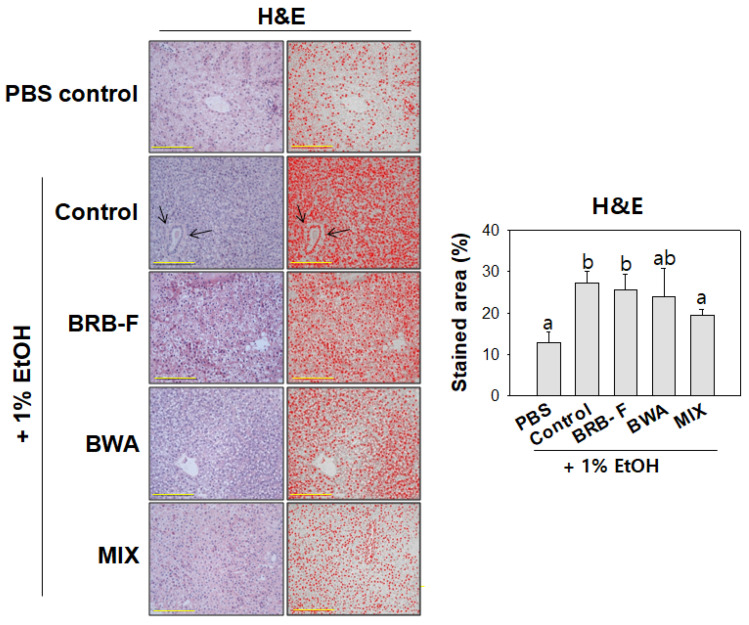
Effect of BRB-F, BWA, and MIX in alcohol liver injury zebrafish for 10 days on the hepatic morphology (400×, Scale bar: 100 μm). Data are the mean ± SEM; ^a,b^ mean values with different superscript letters are significantly different among the groups (*p* < 0.05); PBS, zebrafish exposed to 1% PBS, n = 20; EtOH, zebrafish exposed to 1% EtOH, n = 20; BRB-F, EtOH fed 10% black rice bran, n = 20; BWA, EtOH fed 10% beeswax alcohol, n = 20; MIX, EtOH fed a mixture of BWA and BRB-F (1:6, *w*/*w*), n = 20. H&E-stained transverse section of the liver; representative photomicrographs of the liver are shown at 400× magnification; black arrows indicate alcoholic foamy degeneration; scale bar indicates 100 μm.

**Figure 4 biomolecules-13-00136-f004:**
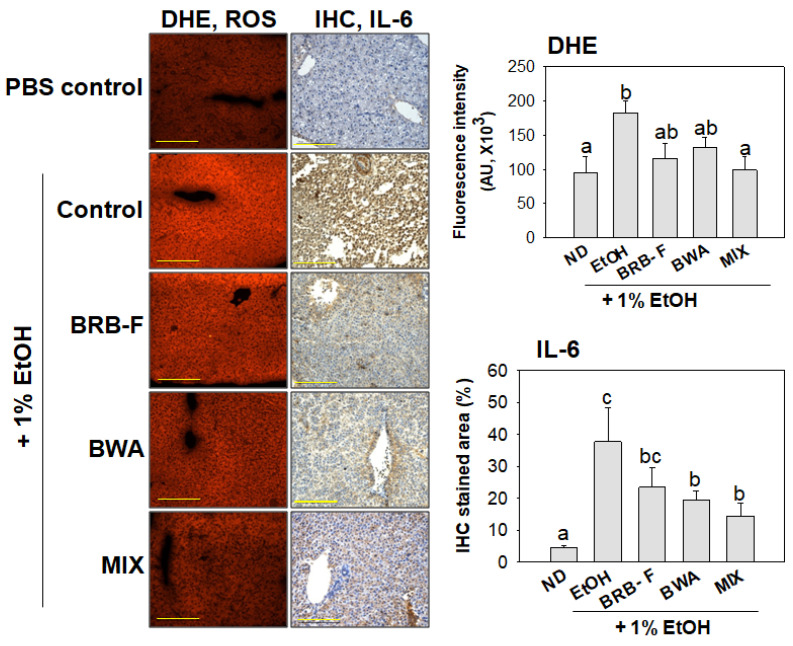
Effect of BRB-F, BWA, and MIX in alcohol liver injury zebrafish for 10 days on hepatic histology for reactive oxygen species production and IL-6 expression (400×, Scale bar: 100 μm). The data are the mean ± SEM; ^a,b,c^ mean values with different superscript letters are significantly different among all groups (*p* < 0.05); PBS, zebrafish exposed to 1% PBS, n = 20; EtOH, zebrafish exposed to 1% ethanol, n = 20; BRB-F, EtOH fed 10% black rice bran, n = 20; BWA, EtOH fed 10% beeswax alcohol, n = 20; MIX, EtOH fed a mixture of BWA and BRB-F (1:6, *w*/*w*), n = 20; DHE, dihydroethidium; IL-6, interleukin-6; IHC, Immunohistochemistry. DHE- and IL-6-stained transverse section of the liver; representative photomicrographs of the liver are shown at 400× magnification. The scale bar indicates 100 μm.

**Figure 5 biomolecules-13-00136-f005:**
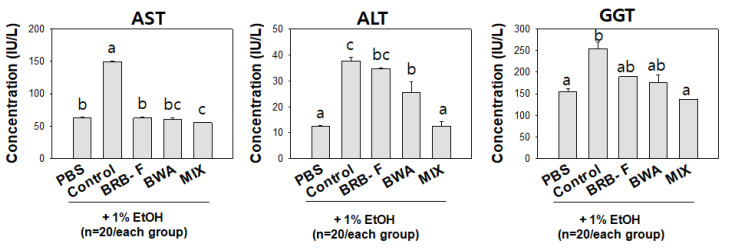
Effect of BRB-F, BWA, and MIX in ethanol-fed zebrafish (n = 20) for 10 days on plasma biomarkers to evaluate the liver function. Data are mean ± SEM; ^a,b,c^ mean values with different superscript letters are significantly different among the groups (*p* < 0.05). PBS, zebrafish exposed to 1% PBS; EtOH, zebrafish exposed to 1% ethanol; BRB-F, EtOH + 10% black rice bran (wt/wt); BWA, EtOH + 10% beeswax alcohol (wt/wt); MIX, EtOH + 10% mixture of BWA and BRB-F(1:6, *w*/*w*); AST aspartate aminotransferase; ALT, alanine aminotransferase; GGT, γ-glutamyl transferase.

**Figure 6 biomolecules-13-00136-f006:**
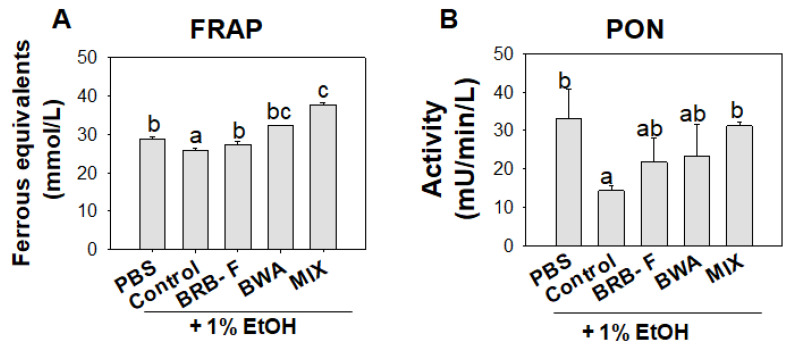
Effect of BRB-F, BWA, and MIX in alcohol liver injury zebrafish for 10 days on the ferric-reducing ability of plasma (**A**) FRAP values and (**B**) paraoxonase activity. The data are mean ± SEM; ^a,b,c^ mean values with different superscript letters are significantly different among all groups (*p* < 0.05). PBS, zebrafish exposed to 1% PBS, n = 20; EtOH, zebrafish exposed to 1% ethanol, n = 20; BRB-F, EtOH fed 10% black rice bran, n = 20; BWA, EtOH fed 10% beeswax alcohol, n = 20; MIX, EtOH fed a mixture of BWA and BRB-F (1:6, *w*/*w*), n = 20; FRAP, ferric-reducing ability of plasma; PON, paraoxonase.

**Figure 7 biomolecules-13-00136-f007:**
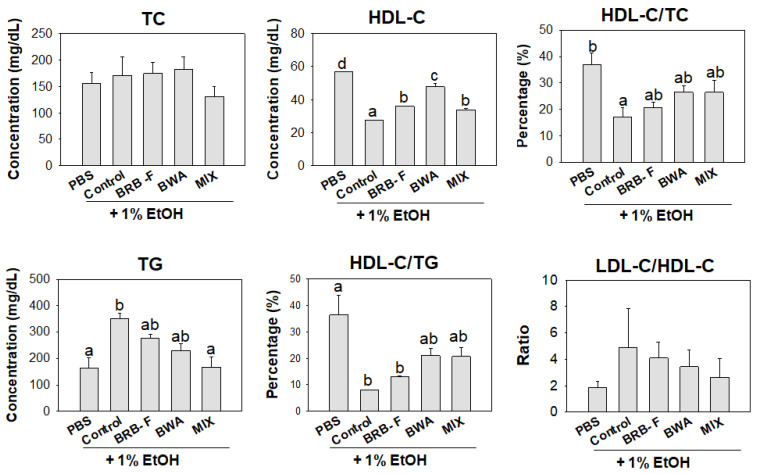
Effect of BRB-F, BWA, and MIX in alcohol liver injury zebrafish for 10 days on the plasma lipid profile. Data are mean ± SEM; ^a,b,c,d^ mean values with different superscript letters are significantly different among the groups (*p* < 0.05). PBS, zebrafish exposed to 1% PBS, n = 20; EtOH, zebrafish exposed to 1% ethanol, n = 20; BRB-F, EtOH fed 10% black rice bran, n = 20; BWA, EtOH fed 10% beeswax alcohol, n = 20; MIX, EtOH fed a mixture of BWA and BRB-F(1:6, *w*/*w*), n = 20; TC, total cholesterol; HDL-C, high-density lipoprotein cholesterol; TG, triglyceride; LDL-C, low-density lipoprotein cholesterol.

**Table 1 biomolecules-13-00136-t001:** Experimental design and changes in body weights during the alcohol exposure period *.

Group	Diet Type	Wet Weight (mg)		
		Day 0 (n = 60)	Day 10 (n = 20)	Day 17 (n = 20)
PBS alone	Normal diet (ND)	316 ± 43	365 ± 49	353 ± 47
EtOH alone	ND	303 ± 43	328 ± 38	N/A
EtOH + BRB-F	ND with BRB-F (10% wt/wt)	310 ± 45	328 ± 64	299 ± 48
EtOH + BWA	ND with BWA (10%, wt/wt)	300 ± 48	304 ± 45	261 ± 51
EtOH + MIX	ND with Mixture of BRB-F and BWA (6:1, wt/wt)	329 ± 39	327 ± 38	287 ± 44

* Data are shown as mean ± SEM. PBS, zebrafish exposed to 1% phosphate buffered saline (PBS); EtOH, zebrafish exposed to 1% ethanol; BRB-F, EtOH fed 10% black rice bran; BWA, EtOH fed 10% beeswax alcohol; MIX, EtOH fed a mixture of BRB-F: BWA (6:1, *w*/*w*); N/A, not applicable due to death.

**Table 2 biomolecules-13-00136-t002:** Effect of BRB-F, BWA, and MIX in alcohol liver injury zebrafish for 17 days on plasma lipid profile and liver toxicity biomarkers.

Group	n	TC (mg/dL)	TG (mg/dL)	AST (IU/L)	ALT (IU/L)	GGT (IU/L)
PBS control	20	155 ± 17 ^ab^	183 ± 5 ^ab^	246 ± 15 ^ab^	109 ± 26	144 ± 12
EtOH + BRB-F	20	182 ± 23 ^b^	304 ± 25 ^b^	316 ± 14 ^b^	144 ± 9	166 ± 3
EtOH + BWA	20	182 ± 23 ^b^	233 ± 49 ^b^	251 ± 5 ^ab^	117 ± 18	162 ± 9
EtOH + MIX (BRB-F: BWA, 6:1)	20	130 ± 19 ^a^	146 ± 8 ^a^	202 ± 11 ^a^	100 ± 8	138 ± 2

Data are mean ± SEM; ^a,b^ mean values with different superscript letters are significantly different among the groups (*p* < 0.05); PBS control, zebrafish exposed to 1% PBS, n = 20; EtOH, zebrafish exposed to 1% ethanol, n = 20; BRB-F, EtOH fed 10% black rice bran, n = 20; BWA, EtOH fed 10% beeswax alcohol, n = 20; MIX, EtOH fed a mixture of BRB-F: BWA (6:1, wt/wt), n = 20; TC, total cholesterol; TG, triglyceride; AST, aspartate aminotransferase; ALT, alanine aminotransferase; GGT, γ-glutamyl transferase.

## Data Availability

The data used to support the findings of this study are available from the corresponding author upon reasonable request.

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
