# Peer review of "Beeswax Alcohol and Fermented Black Rice Bran Synergistically Ameliorated Hepatic Injury and Dyslipidemia to Exert Antioxidant and Anti-Inflammatory Activity in Ethanol-Supplemented Zebrafish"

_biomolecules, 2023, doi:10.3390/biom13010136_

Round 1
Reviewer 1 Report
This manuscript describes the effects of a diet supplemented with beeswax alcohols or fermented black rice brand on zebrafish exposed to 1% ethanol. Results are interesting because zebrafish has been proposed as a model organism to study different liver diseases, including ethanol deleterious effects. However, there are some concerns that hampered the enthusiasm that raises this paper.
Authors shows that beeswax alcohols (BWA) and fermented black rice brand (fBRB) possess antioxidant and anti-inflammatory properties that correlates with ameliorated hepatic injury and dyslipidemia. However, this do not probe a cause-effect relationship. Author must be cautious about this, therefore final conclusions in abstract must be revised: “In conclusion, BWA or BRB showed efficacy to treat alcohol-related metabolic disorders, but the MIX supplement was more effective in ameliorating the liver damage and dyslipidemia via enhanced anti-oxidant and anti-inflammatory activity”. Showed data agree with this hypothesis, but further data/experiments are needed to probe a cause/effect relationship. May be you can write: “In conclusion, BWA or BRB showed efficacy to treat alcohol-related metabolic disorders, but the MIX supplement was more effective in ameliorating the liver damage and dyslipidemia, this last agree with an enhanced anti-oxidant and anti-inflammatory activity exhibited by BWA/BRB”, or something like that.
Introduction is completely centered on effects of heavy alcohol consumption in humans, but you need to explain first why zebrafish is a suitable model for liver disease research. Indeed, observe that zebrafish exposed to 1% ethanol (v/v) is a very high concentration that kill zebrafish after two weeks of treatment (I understand that this is the usual concentration employed with zebrafish, but do not forget that correspond to 171 mM and this concentration never is observed in the blood of heavy intoxicated people with ethanol). In consequence, extrapolation to humans, of observed effects of ethanol in this paper is not straightforward. Readers need help to understand the similarities and differences between ethanol effects in humans, rodents and zebrafish. If you preffer, this last can be explained in the discussion section.
On other hand, observe that the full experimental details must be provided such that the results can be reproduced. For example, how do you feed the animals (one time per day, two times? Observe that the weight of zebrafish in the control group diminished from day 10 to day 17. Why? You mentioned that zebrafish were divided randomly into five groups (n=60). This mean 60 zebrafish for each group (with a total of 300 zebrafish)? How do you obtain plasma samples from zebrafish? Explain.
DPPH radical scavenging activity
You need to complete the experiment using different concentrations of BRB, BWA, MIX, and Vit C, in order to obtain IC50 and compare obtained values with previously published data. You can use also TROLOX as antioxidant control. Thus, a plot of DPPH radical scavenging activity (%) vs concentration of each antioxidant must be included as Figure 1A.
How do you dissolve BRB, and BWA to perform the experiments (DPPH radical scavenging activity and ferric ion reduction ability)?
Table 1.
Please review statistical values. Difference between wet weight of animals at day 0 and 10 (316 +- 43 vs 365 +-49 (SEM) in control group do not seems to be statistically significant. The same can be observed in EtOH alone group and EtOH+MIX group. The same problem with the statistical differences is observed in Figure 2B. Indeed, the observed body weights values for EtOH+BRB and EtOH+BWA in Figure 2A do not correspond to the values indicated in Table 1. Please, review.
Lines 189-190
Observe that the following sentence
“After 10 days of exposure to 1% ethanol, the wet body weights in the ND and EtOH groups were significantly higher than the values of day 0 of this study (Table 1).”
Is not correct. The observed differences are not significant. Description of data in this section is misleading. Please review.
Figure 5
Karmen units are almost obsolete units developed by Dr. Arthur Karmen in 1955. Transaminase activities determined by colorimetric/spectrophotometric techniques should not be reported in spectrophotometric units like Karmen units. Please review. The number of plasma samples in each experimental group is not indicated.
Discussion
Lines 363-382
Observe that you have individual data to test if a positive correlation between blood TG and IL-6 is obtained with your experimental data. Obtained results (plots) can be included as supplementary material. Comparison with published results in humans will be more interesting. Also, you can test correlation between IL-6 and blood TC, or HDL-C.
Lines 388-389
Please review reference 51. Probably the reference is incorrect, or the sentence in lines 388-389 is not correct. Review.
Minor comments
Line 46, change “primary aliphatic acids” by primary aliphatic alcohols”
Figures
Use a different color for each different experimental condition. If you can increase some little the symbols size, this will be better. Observe that there is no additional cost for publishing full color graphics.
Line 243
Change “400â…¹magnification”· to “400â…¹ magnification”
Line 262
Please rewrite, you chose a complicated way to describe obtained results: EtOH exposure diminished 50%(?) PON activity, and this effect is blunted by MIX administration (or something like that).
Line 278
The effect of BRB on HDL-C/TC ratio is not the same that the ratio obtained with BWA and MIX groups. Please rewrite.
Reviewer 2 Report
The author presented a systematic study on the synergetic effect of black rice bran and beeswax on alcohol induced hepatic injury for the first time in the literature. The methods were correctly used and the data could support the main conclusions in this study. However, language problems and careless writing mistakes are easily found in this manuscript, and endeavor should be made to improve the legibility of this paper. Besides, several issues should be addressed before acceptation for publication.
(1) Since the BWA intrinsically contains alcohol, the influence of this alcohol on the results should be discussed.
(2) The synergetic effect between BWA and BRB has been confirmed, the author should give an explain or possible mechanism for the occurrence of the synergetic effect in discussion section.
(3) The conclusion section is too short, and a suggestion for the future research could be presented.
(4) What is the full name for PBS?
(5) What is it meant by the sentence “The 1% ethanol solution (v/v) in the system water tank was changed every 24 hours”. Do you mean the fishes were exposed to 1% ethanol every 24h? the expression is blurred, and the author should rewrite the sentence carefully.
(6) The use of abbreviation in this manuscript is messy. some full names were used over and over again after their abbreviations were defined, and for some full names, their abbreviations were defined for several times. Carefully check the whole manuscript.
(7) The component in the MIX, namely the mass ratio between BWA and BRB, was not consistent in this study, for example, the ratios in table 1, in line 106 and line 311.
(8) Explain in the text, why the DPPH radical scavenging activity of BWA increased at very beginning in figure 1.
(9) Large parts of the graphs have no titles for their vertical axis, which should be added.
Round 2
Reviewer 1 Report
Revised manuscript has been improved, however, but concerns remain:
Lines 372-376
Ethanol is a good substrate for Zn-dependent alcohol dehydrogenases in vertebrates. However, these enzymes have broad, indeed promiscuous, specificities, acting on primary, secondary, and cyclic substrates, even steroids (see for example, Pietruszko, R. (1979). Nonethanol substrates of alcohol dehydrogenase. In Majchrowicz, N. (ed.), Biochemistry and Pharmacology of Ethanol, Vol. 1, Plenum Press, New York, p. 87.). This last is related to the fact that alcohol dehydrogenase have a large hydrophobic substrate binding pocket (see for example, Sirota FL et al., (2021) Functional Classification of Super-Large Families of Enzymes Based on Substrate Binding Pocket Residues for Biocatalysis and Enzyme Engineering Applications. Front. Bioeng. Biotechnol. 9:701120. doi: 10.3389/fbioe.2021.701120), that can handle larger hydroxylated hydrophobic chains, cyclical or steroidal molecules such as bile alcohols, retinol, derivatives of epinephrine, serotonin, dopamine and leukotriene catabolism.
With these previous antecedents you cannot postulate that primary aliphatic alcohols cannot be oxidized by alcohol dehydrogenases. To evaluate the effect of beeswax alcohols on ADH, you need to compare administrated doses. How many moles, or concentration, of ethanol is exposed the zebrafish ADH in comparison to the aliphatic alcohols from beewaxs.
On other hand, observe that in Figure 1 you are not following the complete procedure to report the free radical-scavenging activity (FRSA). You need to calculate it using the formula:
FRSA = [(Ac – As)/Ac] x 100
where Ac is the absorbance of the control and As is the absorbance of the tested sample after 60 min. You need to complete the experiment using additional concentrations of BRB, BWA, and MIX, in order to obtain IC50 and compare obtained values with previously published data. Please review papers reporting free radical-scavenging activity to see how to report your data, and compare your obtained IC50 with published previously data (mainly IC50 for vit C). In my opinion, this last is mandatory.
Indeed, you need to add the following paragraph (that you write) at the end of 2.1 section:
“The BRB, BWA, and MIX powder, each 100 mg of powder, was dissolved in 10 mL of ethanol, with continuous agitation for 12 hr at room temperature. Then, the extract was centrifuged (3000â…¹ g ) to pellet down any debris. The supernatant was diluted with ethanol to get 1, 2, 5, 10 mg/mL and tested for the in vitro assay.”
After, in the Figure 1B you need a control with ethanol alone, because probably the observed effect that you report with BWA is due to ethanol used as solvent. This control is mandatory.
Finally, observe that again data in Table 1 do not agree with data in Figure 2B (the minimal weight reported in Table 1 is observed with BRB-F, but in Figure 2B the minimal weight is observed with BWA), In my opinion this reflect a poor management of data.
Minor comments
Line 87, Change
the material is manufactured described previously
by
the material is manufactured as described previously
Lines 126-128
This paragraph is duplicated in lines 123-125. Please delete one of them.
Author Response
Thank you very much heartily for reviewing and critical comments to improve this paper.
Please see attached doc file.
